# The Correlation of Bile Duct Dilatation in Postmortem Computed Tomography of Lethal Intoxication Cases for Different Drug Types—A Retrospective Study

**DOI:** 10.3390/medsci12040065

**Published:** 2024-11-12

**Authors:** Barbara Fliss, Kirththanan Krishnarajah, Lars Ebert, Cora Wunder, Sabine Franckenberg

**Affiliations:** 1Institute of Legal Medicine, Johannes-Gutenberg University Mainz, 55131 Mainz, Germany; 2Institute of Forensic Medicine, University of Zurich, 8057 Zurich, Switzerlandsabine.franckenberg@usz.ch (S.F.); 3Zurich Forensic Science Institute, 8010 Zurich, Switzerland; lars.ebert@for-zh.ch; 4Diagnostic and Interventional Radiology, University Hospital of Zurich, University of Zurich, 8091 Zurich, Switzerland

**Keywords:** postmortem computed tomography, lethal intoxication, opiates, opioids, bile duct dilatation, CBD

## Abstract

Purpose: To assess (I) whether, in autopsy-proven lethal intoxications with opiates/opioids, a dilatation of the common bile duct (CBD) is still visible in postmortem computed tomography (PMCT) and (II) if a dilatation of the CBD might also be measurable for other substance groups (e.g., stimulants, hypnotics, antipsychotics, etc.). Methods: We retrospectively measured the CBD using PMCT in cases with lethal intoxication (*n* = 125) and as a control group in cases with a negative toxicological analysis (*n* = 88). Intoxicating substances were classified into the subgroups (opiates, opioids, stimulants, hypnotics, antipsychotics, gasses, and others). Significance between the study and control groups was tested with the Mann–Whitney U test, and correlations were examined by using crosstables. Results: There was a statistically significant difference between the CBD diameters in the intoxication group overall, when compared to the CBD diameter in the control group (*p* < 0.001). For both subgroups of “opiates” and “opioids”, there was a strong statistically significant difference between the CBD diameter (being wider) in those groups compared to the control group (both *p* = 0.001). For the three subgroups “hypnotics”, “stimulants”, and “psychotropic drugs”, there was no statistically significant difference between the CBD diameters in the intoxication subgroups when compared with the control group. The other subgroups were too small for statistical analysis. Conclusion: A dilated common bile duct in postmortem computed tomography might be used as an indication for a lethal opioid or opiate intoxication only in regard to the specific case circumstances or together with other indicative findings in a postmortem investigation.

## 1. Introduction

Morphine is known to constrict the sphincter of Oddi (SO), which increases biliary pressure within the common bile duct [1,2,3,4]. This effect is utilized in cholangiopancreatography in Magnetic Resonance Imaging (MRI), where intravenous morphine is administered prior to the examination to improve image quality by distention of the biliary and pancreatic ducts [5]. In clinical radiology, in general, this effect of morphine on the biliary duct system can be helpful for the evaluation of patients with primary sclerosing cholangitis or malignant biliary and pancreatic neoplasms [5,6]. The average duration of this effect (morphine as an i.v. medication) usually lasts up to a couple of minutes [7]. The common bile duct (CBD) arises from the union of the common hepatic duct (consisting of the right and left hepatic ducts) with the cystic duct. The CBD is the main duct of the liver and gallbladder and opens into the lumen of the duodenum at the papilla duodenal major. In about 80% of cases, this occurs together with the pancreatic duct, which carries the secretion of the exocrine pancreas. The emptying of the bile into the duodenum is carried out by the SO, which is located at the duodenal junction. It consists of smooth muscle surrounding the bile duct [8]. The diameter of the common bile duct of up to 5–9 mm in ultrasound examinations [9] and of up to 6–8 mm in computed tomography (CT) is considered normal [10]. Thus far, there are no defined normal ranges of the CBD in postmortem CT.

Concerning the influence of other drugs on the SO, it has been shown in an animal model that acetylcholine and prostigmin as well as alcohol increase the electrical activity in the sphincter, whereas atropine showed a depressant effect and amyl nitrite reduced the electrical activity of the sphincter [11]. Also, in the human SO, contraction and pressure declined after the administration of anisodamine, atropine, or scopolaminbutylbromid [4,12].

So far, we do not know if this effect of the opioids or other drugs on the SO persists after death. If that would be the case, a dilatation of the CBD (caused by a constricted SO) might, for example, be used as an indicator for possible intoxication (in the absence of other obvious causes and depending on the forensic context). The common unspecific signs of an intoxication recognized during autopsy like edema of the brain and the lung as well as a distension of the urinary bladder can be reproduced in postmortem CT [13].

Therefore, we want to investigate (I) whether in autopsy-proven lethal intoxications with opiates/opioids, a dilatation of the CBD is still visible in postmortem computed tomography, and (II) if a dilatation of the CBD might also be measurable for other substance groups (e.g., stimulants, hypnotics, antipsychotics, etc.).

## 2. Materials and Methods

### 2.1. Study Group

We searched the archive of the Institute of Forensic Medicne Zurich for all autopsy cases for a three-year period from January 2016 until December 2018. Toxicological information was taken from the autopsy and the associated toxicology report, resulting in 1410 cases. In all cases, an autopsy was ordered by the prosecutor’s office. Before the autopsy, a postmortem CT scan was conducted in each case, which was interpreted by a trained forensic pathologist or a radiologist. The toxicological examinations were carried out upon separate request from the authorities. The results of the examinations were stored in the institute’s database.

We excluded all cases without toxicological analysis, with non-lethal intoxications and an age < 18 years. Also, cases were excluded where a disease was diagnosed that could influence the diameter of the CBD (e.g., pancreatic cancer and all visible alterations, as well as cases known from medical history, with a possible influence on the biliary system). Furthermore, all corpses were excluded that showed putrefaction gas in the biliary system or other signs of decomposition. In forensic routine casework, the exact time of death is typically unknown. Usually, we only have an estimated time of death to work with. Therefore, we decided to exclude all corpses that either exhibited signs of decomposition in the external examination or in postmortem computed tomography.

Overall, we included 125 cases. Of the 125 individuals, 81 were male and 44 were female. The mean age was 45 years, the median age was 43 years, and the age range was 18–99 years. As a control group, we included all cases that had a negative toxicology report (no detection of any toxicological substance such as alcohol, medicines, or drugs) (*n* = 88; males, *n* = 60; females, *n* = 28; mean age = 48 years; median age = 48 years; and age range = 18–88 years).

The control group consisted of 88 cases with a negative toxicology report. Among these cases, 60 were male and 28 were female. The males were aged between 18 and 83 years (median age: 47 years) and the females between 19 and 88 years (median age: 53 years).

### 2.2. Toxicological Analysis

Analysis was routinely performed directly following the autopsy. First, urine was screened by a cloned enzyme donor immunoassay (CEDIA^®^) for opiates, cocaine, cannabis, amphetamines, methadone, barbiturates, benzodiazepines, and lysergic acid diethylamide (LSD), followed by untargeted liquid chromatography tandem mass spectrometry (LC–MS/MS) ion trap screening after dilution and filtration (Bruker amazon^®^, Maurer/Wissenbach/Weber database [14]). Peripheral blood was analyzed for ethanol and other volatile compounds by headspace gas chromatography flame ionization detection (HS-GC-FID). Targeted quantitative analysis of the toxicologically relevant substances was performed using validated LC–MS/MS methods.

### 2.3. Substances

Intoxicating substances were classified into the following substance categories (Table 1): opiates (heroin/morphine, codeine), opioids (methadone, fentanyl, tramadol), stimulants (cocaine, amphetamine and derivatives), hypnotics (benzodiazepines, Z-drugs, barbiturates, antihistamines), antipsychotics (antidepressants, neuroleptics), gasses (CO, helium), and others (chloroquine, acetaminophen, cyanide).

### 2.4. Imaging and Readout

All our cases had undergone postmortem computed tomography (PMCT) before autopsy. PMCT was performed with a 128-slice scanner (SOMATOM Definition Flash, Siemens Healthineers, Erlangen, Germany) with bodies in the supine position using automatic dose modulation (CARE Dose 4D™, Siemens Healthineers, Erlangen, Germany). The imaging parameters included tube voltage, 120 kVp, and slice collimation, 128 × 0.6 mm. PMCT data were reviewed on a Syngo system imaging software VB40 for multimodality reading (Syngo. via, Siemens Healthineers, Erlangen, Germany). A medical student under the supervision of both a board-certified radiologist and a forensic pathologist performed a blinded read-out for these PMCT data (soft kernel reconstruction B30s, soft tissue window), namely, a measurement of the CBD diameter and an evaluation of the gallbladder (present versus post-cholecystectomy) and summarizing the data in an excel sheet (Figure 1). Additionally, the dataset was assessed for the presence of decomposition gas. Cases in which decomposition gas were found in the vascular system or organs were excluded.

### 2.5. Statistics

Continuous variables were examined for normality by visual analysis. The distribution of age and sex were listed as minimal, maximal, and median values.

For testing the significance between the study and control groups, the Mann–Whitney U test was performed using SPSS (Version 27). A *p*-value < 0.001 was considered statistically significant. The r-value was calculated manually and interpreted according to Cohen [15] (see below).

The correlations were examined by using crosstables (Pearson and eta correlation) and interpreted according to Cohen. Absolute values of the correlation coefficient r = 0.1–0.3 were regarded as a weak, r = 0.3–0.5 as a moderate, and r = 0.5–1.0 as a strong correlation.

### 2.6. Ethics

This research project does not fall within the scope of the Human Research Act (HRA). Therefore, authorization from the ethics committee is not required (KEK ZH-Nr. 15-0686).

## 3. Results

In 62% of the cases with lethal intoxications, one substance alone or several substances from the same substance group were detected. The remaining 38% of the cases showed a mixture of substances from different substance groups. To further simplify the analysis, the substance most likely to be fatal was considered the leading intoxicant in those mixed intoxication cases and defined the assignment of the case to the corresponding substance group. The other substances of the case were considered as having less relevance for death.

Since the intoxication subgroups “gasses” and “others” contained only five and six cases, respectively, they were excluded from the statistical analysis.

The distribution of CBD diameters in both groups is shown in Figure 2. The diameter ranged between 2 and 11 mm (median diameter: 5 mm) in the study group (Figure 2a) and between 2 and 8 mm (median diameter: 4 mm) in the control group (Figure 2b). In the control group, the diameter of the CBD ranged in males between 2 and 8 mm (median diameter: 4 mm) and in females between 2 and 7 mm (median diameter: 3 mm).

A diameter of >8 mm was considered pathological, similar to the known clinical reference values [10].

The Mann–Whitney U test showed a statistically significant difference between the CBD diameters in the intoxication group overall, when compared to the CBD diameters in the control group (*p* < 0.001; r = 0.23).

There was a weak correlation between the CBD diameter and sex (study group, r = 0.066, *p* = 0.462; control group, r = 0.244, *p* = 0.022), with slightly larger CBD diameters observed in males (Figure 3).

We also found a weak correlation between age and the CBD diameter (study group, r = 0.28, *p* = 0.754; control group, r = 0.11, *p* = 0.916).

For both subgroups of “opiates” and “opioids”, there was a strong statistically significant difference between the CBD diameter (being wider) in those groups compared to that in the control group (both *p* = 0.001) (Figure 4). Diameters > 8 mm were found in both subgroups, whereas only one CBD with a diameter of >8 mm was found in the psychotropic drugs subgroup, and none in the control group. For the other three subgroups, there was no statistically significant difference between the CBD diameter in the intoxication subgroups compared with that in the control group (stimulants, *p* = 0.462, r = 0.039; hypnotics, *p* = 0.244, r = 0.161; psychotropic drugs, *p* = 0.142, r = 0.299) (Figure 4).

## 4. Discussion

The constricting effect of opioids or opiates on the SO is well known in a clinical setting. We hypothesized that, in autopsy-proven lethal intoxications with opiates/opioids, a dilatation of the CBD is still visible in postmortem computed tomography and (II) a dilatation of the CBD might also be measurable for other substance groups (e.g., stimulants, hypnotics, antipsychotics, etc.).

The first aim of our study was to investigate whether, in autopsy-proven lethal intoxications with opiates or opioids, a dilatation of the CBD is still visible in postmortem computed tomography.

We could indeed show that a dilatation of the CBD (>8 mm) was significantly correlated with lethal intoxication in opioid or opiate cases, which is already known in clinical diagnostics [10]. This means that the effect of opioids and opiates persists after death. Therefore, a dilated CBD might be used as an indicator for intoxication in postmortem investigations. Certainly, this finding should be regarded carefully and only in the context of the case circumstances. Nevertheless, it might act as an additional indication for a possible lethal intoxication, alongside other already established signs of intoxication, such as a full urinary bladder, brain edema, or lung edema [13,16]. The correlation between age/sex and the dilatation of the CBD was weak, so it should not be taken into account.

A limitation of our study is that we did not regard the possible influence of the time of death interval on the outcome. It might be interesting to repeat our study on defined time of death intervals to evaluate how long exactly the dilatation of the CBD persists postmortem. Our finding of a dilated CBD in lethal opioid or opiate intoxications might be even more pronounced in only recently deceased bodies.

The second aim of our study was to evaluate if a dilatation of the CBD might also be seen in lethal intoxication for other subgroups of drugs.

In subgroups with large enough case numbers (such as “stimulants”, “hypnotics”, and “antipsychotics”), we found no correlation between lethal intoxication and the dilatation of the CBD. As a limitation for this subgroup analysis, the case number for some of the subgroups (for example, “gasses”) was unfortunately too low for statistical analysis. Further studies with larger case numbers are needed.

Normal ranges for the CBD are not known for postmortem CT. In clinical patients, a CBD diameter range of up to 6–8 mm in computed tomography (CT) is specified as normal [10]. We found a statistical significance between an CBD diameter of >8 mm and opioid/opiate intoxication.

In our study, we observed diameters of 2–8 mm in the control group (median diameter of 3 mm in females and 4 mm in males). Although the control group consisted of only 88 individuals, it seems that the CBD diameter is smaller in corpses.

## 5. Conclusions

A dilated common bile duct in postmortem computed tomography might be used as an indication for a lethal opioid or opiate intoxication, and a statistical significance between a CBD of >8 mm and opioid/opiate intoxication was found, but only in specific case circumstances or together with other indicative findings in a postmortem investigation. As a singular finding, it should be interpreted with great caution because a normal diameter of the CBD does not exclude intoxication.

## Figures and Tables

**Figure 1 medsci-12-00065-f001:**
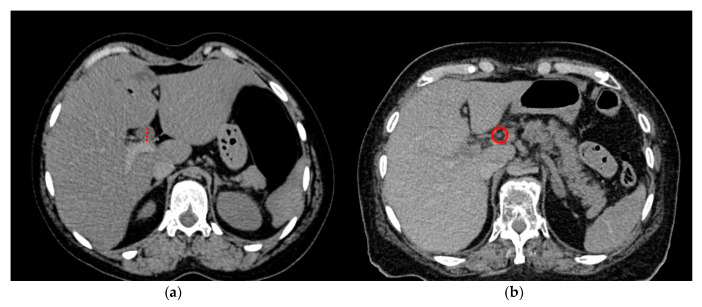
Examples of the CBD measurement in PMCT on the axial plane in a soft tissue window. (**a**) Significantly dilated CBD (14 mm, red dotted line) in a case of lethal opioid intoxication and (**b**) normal diameter (3 mm, red circle) of the CBD in a control group case.

**Figure 2 medsci-12-00065-f002:**
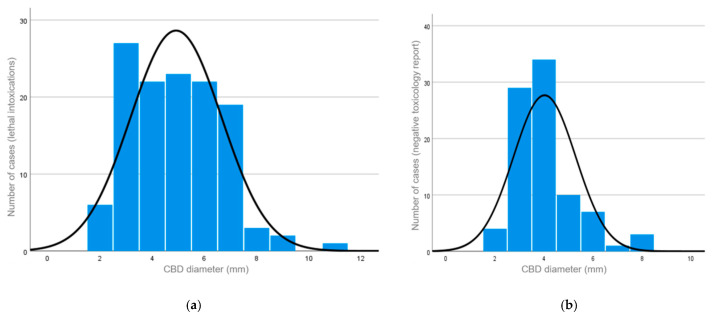
(**a**) The distribution of the CBD diameters for the study group with lethal intoxications (median = 5; standard deviation = 1.741; *n* = 125). (**b**) The distribution of the CBD diameters for the control group with a negative toxicology report (median = 4; standard deviation = 1.268; *n* = 88).

**Figure 3 medsci-12-00065-f003:**
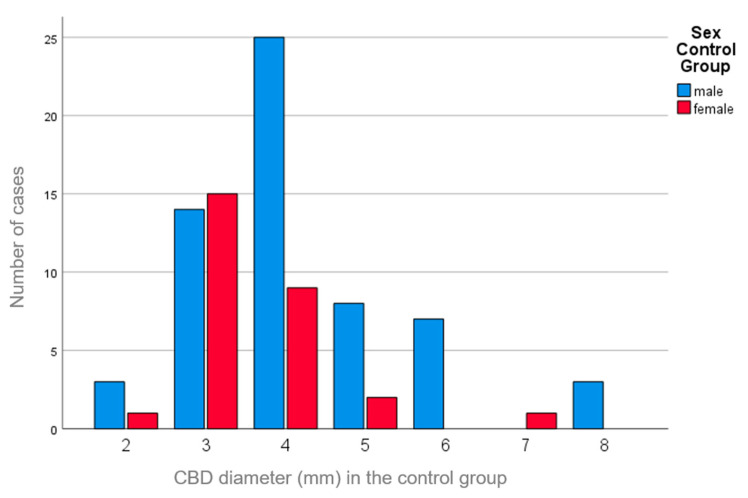
Bar graph of the sex difference in the distribution of the CBD diameters for the control group with negative intoxication; blue = male; red = female.

**Figure 4 medsci-12-00065-f004:**
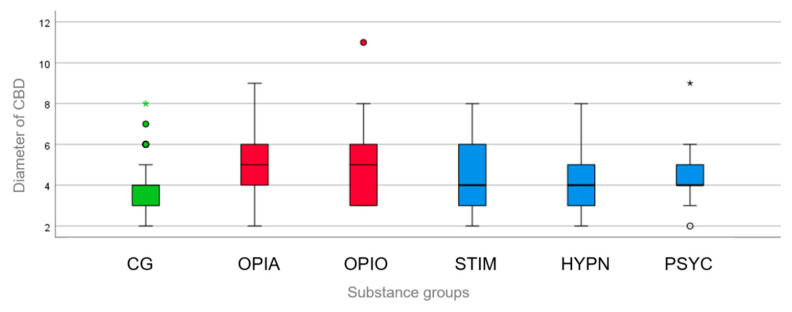
Boxplot for the CBD diameters in the control group (CG, green) and the different subgroups of “opiates” (OPIA), “opioids” (OPIO), “stimulants” (STIM), “hypnotics” (HYPN), and “psychotropic drugs” (PSYC). Only in the subgroups of “opiates” and “opioids” (red bars), the CBD was significantly wider than that in the control group. There was no such correlation for the other subgroups (blue bars). Outliers that differed significantly from the rest of the dataset are plotted as individual points or asterisks beyond the whiskers on the boxplot.

**Table 1 medsci-12-00065-t001:** Distribution of cases for the different substance groups. The substance groups “gasses” and “others” were not regarded further due to their low number of cases.

Scheme 49	*n*	Diameter CBD in mm		
		Median	Minimum	Maximum
Opiates	49	5	2	9
Opioids	42	5	3	11
Stimulants	9	4	2	8
Hypnotics	19	4	2	8
Psychotropic drugs	13	4	2	9
Gasses	5	7	3	7
Others	6	6	4	7

## Data Availability

Data are available in Table 1 and Figure 1, Figure 2, Figure 3 and Figure 4.

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
