# Peer review of "The Correlation of Bile Duct Dilatation in Postmortem Computed Tomography of Lethal Intoxication Cases for Different Drug Types—A Retrospective Study"

_medsci, 2024, doi:10.3390/medsci12040065_

Round 1
Reviewer 1 Report
Comments and Suggestions for Authors
Line 35 – changed “increased” to “increases”
Line 36 – remove “for example”
Line 53 – change “it could be shown for example” to “it has been shown”
Line 57 – remove “for example”
Line 61 – remove “for example”
Line 76 – change “hose” to “those”
Materials and Methods:
This section would be strengthened by adding more information about your samples. Of the 125 cases, how many were male versus female? What is the ancestry? What was the age range? *Note I found some of this information in the Results Section. It should be moved to Materials and Methods.
Hoe long after death was PMCT conducted? *Note I did see you addressed this in the Discussion
ANCOVA may be a better statistical test to look for associations between demographic variables (including interactions) and measurement findings? Variables would include the age, sex, ancestry, sex/population interactions, diameter and substance group.
General:
Overall, the article is nicely written, and the figures supplement the written information. The abstract is a nice summary of the paper. The keywords are appropriate. The Introduction sufficiently introduces the topic. I had a few questions and suggestion for the Material and Methods, but they were minor. The Results were clear, and the Discussion was thorough.
Author Response
“Correlation of Bile Duct Dilatation in Postmortem Computed Tomography of Lethal Intoxication Cases for Different Drug Types - A Retrospective Study”
Dear Reviewer,
Thank you very much for your constructive comments, which really helped to improve the paper. The comments are addressed as below. The typos were of course corrected.
Materials and Methods:
This section would be strengthened by adding more information about your samples. Of the 125 cases, how many were male versus female? What is the ancestry? What was the age range? *Note I found some of this information in the Results Section. It should be moved to Materials and Methods.
Answer: The information is moved to Materials and Methods.
Hoe long after death was PMCT conducted? *Note I did see you addressed this in the Discussion
Answer: We added it also to the Material and Method Section.
ANCOVA may be a better statistical test to look for associations between demographic variables (including interactions) and measurement findings? Variables would include the age, sex, ancestry, sex/population interactions, diameter and substance group.
Answer: Thank you very much for your suggestion. We preferred the Mann-Whitney U test over ANCOVA because for small sample sizes, the Mann-Whitney U test provides more reliable results, especially when the distribution is not normal, while ANCOVA may struggle with small samples.
General:
Overall, the article is nicely written, and the figures supplement the written information. The abstract is a nice summary of the paper. The keywords are appropriate. The Introduction sufficiently introduces the topic. I had a few questions and suggestion for the Material and Methods, but they were minor. The Results were clear, and the Discussion was thorough.
Answer: Thank you very much for your kind comments.
Reviewer 2 Report
Comments and Suggestions for Authors
Dear authours, I was asked to review your paper dealing with bile duct dilatation and lethal intoxication. I think the paper is well written. And the approach is quite innovative. For me, there are just some specifications needed and some information is missing. Here are my propositions: 1. Line 35: it should be « increases » instead of « increased ». 2. Mat & Meth: I propose to start with an introducing part of this chapter explaining how you work in your center: Tell that in all cases where an autopsy is demanded, you perform a PMCT, that all PMCTs are interpreted by….that you keep all auotpsy reports and all radilogical reports in a data base, that toxicological exams are performend on special request of the prosecuter…. This would be a help to better understand the rest of this chapter. 3. Mat & Meth: You should explain wich information was exctrated from wich report and by whom. How did you report the information (Excel spread sheet, special software?) 4. You say that you excluded cases « showing putrefaction gas in the bilary system ». But how did you get this information? I mean this is normally not discribed in autopsy reports or in radiological reports. Did you do a specific reading ? Who did it? Should this not be explained in the chapter 2.4? 5. Why did you not report the time between death and PMCT. Don’t you have this information ? If Not, you have to explain it. However, this is a pity. And at least you should discribe the State of the body (indicating if there were no signs of putrefaction, early signs of putrefaction (green spot on the belly…..) This information can easily be extracted from autopsy reports. 6. You should add the radiological alteration index of the cases, as another indicator of the state of the body. 7. Line 76: it should be « those » not « hose ». 8. Line 127:there are two « , » between the words « individuals » and « in ».
9. I think it would be of great interest to know more about the control cases. You could add a table showing this information (including cause auf death, age, sex etc.)
10. Discussion: This discussion is missing a reflexion about the pathophysiology. How do you explain your results? Are they surprising to you (Also concerning the control group)? Did you await them? What may they suggest… You must have some explanations and hyothesis. Why do you not discuss them? This would be the most interresting part of the paper.
11. Lines 191-195: The missing information about post-mortem Interval is indeed the biggest limitation of your study. I can understand that it is not possible to completely overcome this, but if you add a table indicating eventual early signs of alteration, other elements of body state (type of rigidity, absence of rigidity, etc.) and the RAI, you can at least give a partial indication. I absolutely think that this is necessaray (see my comments above).
12. Line 206: 3rd word: I think it should be « Although » instead of « Also ».
13. Discussion and conclusion: In your paper a clear statement concerning the final result is missing. In your conclusion, you say that « A dilated commun bile duct in postmortem computed tomography might be used as an indication for a lethal opioid or opiate intoxication… ». But finally you do not define what is a « dilation » in the PMCT. What is the proposed cut-off value? More than 8 mm? As you have this diameter in your intoxication cases as well as in your control group, it is not clear for me. You should discuss this in the « Discussion » and give the information as a result in the chapter « Results » and « Conclusion ». Otherwise, your paper is not really useful.
Author Response
“Correlation of Bile Duct Dilatation in Postmortem Computed Tomography of Lethal Intoxication Cases for Different Drug Types - A Retrospective Study”
Dear Reviewer,
Thank you very much for your constructive comments, which really helped to improve the paper. The comments are addressed as below. The typos were of course corrected.
Mat & Meth: I propose to start with an introducing part of this chapter explaining how you work in your center: Tell that in all cases where an autopsy is demanded, you perform a PMCT, that all PMCTs are interpreted by….that you keep all auotpsy reports and all radilogical reports in a data base, that toxicological exams are performend on special request of the prosecuter…. This would be a help to better understand the rest of this chapter.
Answer: Thank you for this comment. We included the information.- Mat & Meth: You should explain wich information was exctrated from wich report and by whom. How did you report the information (Excel spread sheet, special software?)
Answer: Thank you for this comment. We included the information. - You say that you excluded cases « showing putrefaction gas in the bilary system ». But how did you get this information? I mean this is normally not discribed in autopsy reports or in radiological reports. Did you do a specific reading ? Who did it? Should this not be explained in the chapter 2.4?
Answer: Thank you for this comment. We included the information.
- Why did you not report the time between death and PMCT. Don’t you have this information ? If Not, you have to explain it. However, this is a pity. And at least you should discribe the State of the body (indicating if there were no signs of putrefaction, early signs of putrefaction (green spot on the belly…..) This information can easily be extracted from autopsy reports.
Answer: In forensic routine casework, the exact time of death is typically unknown. Usually, we only have an estimated time of death to work with. Therefore, we decided to exclude all corpses that either exhibited signs of decomposition in the external examination or on post-mortem computed tomography. - You should add the radiological alteration index of the cases, as another indicator of the state of the body.
Answer: Thank you for the suggestion. We only included corpses with no gas in the organs and the vessels as well as with no external signs of decomposition.
- Line 76: it should be « those » not « hose ».
Answer: Corrected. - Line 127:there are two « , » between the words « individuals » and « in ».
Answer: Corrected.
- I think it would be of great interest to know more about the control cases. You could add a table showing this information (including cause auf death, age, sex etc.)
Answer: The demographical data have been added. - Discussion: This discussion is missing a reflexion about the pathophysiology. How do you explain your results? Are they surprising to you (Also concerning the control group)? Did you await them? What may they suggest… You must have some explanations and hyothesis. Why do you not discuss them? This would be the most interresting part of the paper.
Answer: Thank we added the information you asked for.
- Lines 191-195: The missing information about post-mortem Interval is indeed the biggest limitation of your study. I can understand that it is not possible to completely overcome this, but if you add a table indicating eventual early signs of alteration, other elements of body state (type of rigidity, absence of rigidity, etc.) and the RAI, you can at least give a partial indication. I absolutely think that this is necessaray (see my comments above).
Answer: Thank you for the suggestion. We only included corpses with no gas in the organs and the vessels as well as with no external signs of decomposition. - Line 206: 3rd word: I think it should be « Although » instead of « Also ».
Answer: Corrected. - Discussion and conclusion: In your paper a clear statement concerning the final result is missing. In your conclusion, you say that « A dilated commun bile duct in postmortem computed tomography might be used as an indication for a lethal opioid or opiate intoxication… ». But finally you do not define what is a « dilation » in the PMCT. What is the proposed cut-off value? More than 8 mm? As you have this diameter in your intoxication cases as well as in your control group, it is not clear for me. You should discuss this in the « Discussion » and give the information as a result in the chapter « Results » and « Conclusion ». Otherwise, your paper is not really useful.
Answer: Thank you, the information have been added.